# Prenatal versus Postnatal Initial Colonization of Healthy Neonates' Colon Ecosystem by the Enterobacterium *Escherichia coli*

Mohammad Al-Balawi,[a] ⓘ Fatthy Mohamed Morsy[a,b]

[a]Biology Department, Faculty of Science, Taibah University, Medina, Saudi Arabia
[b]Bacteriology Section, Botany and Microbiology Department, Faculty of Science, Assiut University, Assiut, Egypt

**ABSTRACT** The human colon is a microbial ecosystem whose initial bacterial colonization in neonates is an important step in establishing a beneficial microbiota for the body's health. This study investigated the occurrence of viable culturable *Escherichia coli* in first-day meconium versus subsequent days' stool to explore the prenatal versus postnatal initial colonization of the colon by *E. coli* in healthy neonates. *E. coli* occurrence was investigated on eosin-methylene blue (EMB) agar, followed by morphological and biochemical characterizations and phylogenetic analysis of 16S rRNA-encoding gene sequences. Viable culturable *E. coli* was not detected in meconium of healthy male or female neonates delivered either vaginally or by cesarean section. Neonates delivered surgically also showed no *E. coli* colonization on the second and third days, confirming postnatal colonization of the colon by this enterobacterium. *E. coli*'s initial colonization in the colon of neonates delivered vaginally occurred on the second day, which can be attributed to inoculation from the vaginal canal during delivery and, in comparison to the colonization in neonates delivered surgically, leads to the inference that the bacterium is not originally found in meconium. This study suggests no viability of the meconium microbiome in healthy neonates, possibly due to antimicrobial action in the prenatal colon's meconium protecting babies' gut from infection during delivery.

**IMPORTANCE** The results of this study suggest that the initial postnatal colonization of neonates' colon by beneficial bacteria is a naturally controlled process in which the prenatal colon's meconium might play a role in protecting against infection of the babies' gut during delivery.

**KEYWORDS** *Escherichia coli*, colon bacteria, healthy neonates, initial colonization, meconium

The human colon as a microbial ecosystem retains a microbial community playing important roles in the body's health (1–3). The initial colonization of the colon by bacteria is important for formulating the microbial community; however, this initial bacterial colonization is a complicated process that occurs in correlation with some factors that might affect the establishment of the microbiome within the colon of neonates (4–6). The origin of the bacterial inoculation for the establishment of the colon microbiome is mostly expected to be postnatal, while the prenatal colon is sterile. However, bacterial DNA was detected in the first-pass meconium and a meconium microbiome was reasonably reported (7–13). Thus, the colonization of the human gut was suggested to initiate prenatally *in utero*, where inoculation might reasonably occur from distinct microbial communities in the placenta and amniotic fluid (14, 15), since a human uterine microbiome was reported even in nonpregnant women (16). These reports are largely based on direct DNA sample extraction and subsequent molecular

Address correspondence to Fatthy Mohamed Morsy, fmorsy@aun.edu.eg.

Microbiology
Spectrum

biological identifications and analysis. Despite the detection of some bacteria in a culture-dependent approach, the microbiomes of the placenta, amniotic fluid, and meconium studied by molecular biological tools were concluded to have low richness and diversity (14).

*Escherichia coli* is a basic member of the colon microbial community (17, 18) and an indicator of the fecal microbiota (19). In spite of its frequent pathogenicity, *E. coli*, as one of the normal microbiota in the human gut, plays several important roles in human health (20). *E. coli* was suggested to play an important role in sustaining the fermentation processes in the colon. Fermentation occurs in the human colon, which retains hundreds of different bacterial species (5, 6, 21), and the fermentation processes and the bacterial colonization of the colon are of importance to human health in many aspects (5, 6). The relationships between different components of the colon bacteria are not fully understood. The fermentative microaerophilic and anaerobic bacteria found in the colon would require the establishment and maintenance of anaerobiosis within the colon. Microaerophilic and anaerobic bacteria colonize the neonate's large intestine (22, 23), which requires the establishment of anaerobic conditions favored by such health-beneficial bacteria. The occurrence of *E. coli* in the colon at the early stage of human life might reveal its importance for the body's development after birth and a possible way by which the establishment of anaerobiosis occurs, thereby setting up favorable conditions for colonization by microaerophilic and anaerobic health-beneficial bacteria. *E. coli*, as a facultative anaerobe that can consume molecular oxygen, can help in establishing the microaerobic or anaerobic conditions required for beneficial microaerobic and anaerobic bacteria in the colon, such as *Lactobacillus* spp. and others. While colonization of the human colon by *E. coli* and other bacteria varies according to the mode of delivery (24), type of feeding, and several other factors (25–28), the recently proposed prenatal versus postnatal initial colonization should be explored for this fundamental bacterial member of the human colon microbiome. In this study, we followed the occurrence of viable culturable *E. coli* as a basic member of the *Enterobacteriaceae* in the first-day meconium and subsequent days' stool samples to follow up its viable initial colonization of the colon in healthy neonates.

## RESULTS

**Prenatal versus postnatal initial colonization of the colon by *E. coli* in healthy neonates.** In this study, the postnatal versus prenatal initial colonization of the colon by the enterobacterium *E. coli* was followed to examine its occurrence in the first-pass meconium and subsequent days' stool of healthy newborn babies. *E. coli* was not detected in the first-day meconium of any of the newborn babies investigated in this study (Table 1). *E. coli* was not detected at all in the second- and third-day stool samples of any of the newborn babies delivered by cesarean section, either in male or female babies (Table 1), while its occurrence initiated at the fourth day (Fig. 1). These results indicate that *E. coli* colonization of the colon is postnatal, as clearly shown in the case of newborn male and female babies delivered by cesarean section. While the occurrence of *E. coli* was initiated in the second-day stool of vaginally delivered male and female babies (Fig. 1), it was not present in 20% to 60% of babies up to the fourth day (Table 1), whereas subsequently, at the 5th day, all neonates retained the bacterium. The absence of *E. coli* in meconium and its appearance in only 20% to 60% of vaginally delivered babies do not support a prenatal inoculation but confirm the same conclusion of a postnatal initial colonization that clearly appeared in the case of neonates delivered by cesarean section. For a confirmed identification, phylogenetic analysis of 16S rRNA-encoding genes was conducted for *E. coli* strains isolated from male neonates born vaginally (Fig. 2), female neonates born vaginally (Fig. 3), male neonates born by cesarean section (Fig. 4), and female neonates born by cesarean section (Fig. 5). All strains were identified as *E. coli* by phylogenetic analysis of the 16S rRNA-encoding gene sequences (Fig. 2 to 5).

**Postnatal initial colonization of colon by *E. coli* in healthy newborn babies in the first week of their life is natural and occurs universally, whatever the mode of delivery or sex.** Statistical analysis of the *E. coli* CFU among meconium or stool samples of male and female newborn babies delivered vaginally or by cesarean section at

**TABLE 1** CFU counts of *Escherichia coli* in first-day meconium and subsequent days' stool from neonates in the first week of life

| Day of age | Neonate subject no. | CFU/g stool ($\times 10^6$) from neonate delivered as indicated[a] | | | |
|---|---|---|---|---|---|
| | | Vaginal birth | | Caesarean section | |
| | | Male | Female | Male | Female |
| 1 (first-pass meconium) | 1 | ND | ND | ND | ND |
| | 2 | ND | ND | ND | ND |
| | 3 | ND | ND | ND | ND |
| | 4 | ND | ND | ND | ND |
| | 5 | ND | ND | ND | ND |
| 2 | 1 | ND | ND | ND | ND |
| | 2 | 8 | 8 | ND | ND |
| | 3 | ND | ND | ND | ND |
| | 4 | 33 | 14 | ND | ND |
| | 5 | 29 | 32 | ND | ND |
| | Mean | 14 | 10.8 | 0 | 0 |
| | SE | 7.1 | 5.9 | 0 | 0 |
| 3 | 1 | ND | ND | ND | ND |
| | 2 | 4 | ND | ND | ND |
| | 3 | 51 | 9 | ND | ND |
| | 4 | 17 | 18 | ND | ND |
| | 5 | 9 | ND | ND | ND |
| | Mean | 16.2 | 5.4 | 0 | 0 |
| | SE | 9.2 | 3.6 | 0 | 0 |
| 4 | 1 | ND | ND | ND | 11 |
| | 2 | ND | 35 | 20 | ND |
| | 3 | ND | 22 | 31 | ND |
| | 4 | 28 | 13 | 39 | 49 |
| | 5 | 14 | 25 | 14 | 26 |
| | Mean | 8.4 | 19 | 20.8 | 17.2 |
| | SE | 5.6 | 5.9 | 6.8 | 9.3 |
| 5 | 1 | 32 | 24 | 32 | 37 |
| | 2 | 41 | 38 | 34 | 53 |
| | 3 | 53 | 49 | 46 | 38 |
| | 4 | 48 | 51 | 38 | 47 |
| | 5 | 46 | 48 | 42 | 43 |
| | Mean | 44 | 42 | 38.4 | 43.6 |
| | SE | 3.6 | 5 | 2.6 | 3 |
| 6 | 1 | 41 | 41 | 51 | 29 |
| | 2 | 62 | 54 | 67 | 45 |
| | 3 | 68 | 59 | 44 | 59 |
| | 4 | 53 | 50 | 47 | 49 |
| | 5 | 59 | 63 | 56 | 63 |
| | Mean | 56.6 | 53.4 | 53 | 49 |
| | SE | 4.6 | 3.8 | 4 | 6 |

[a]ND, not detected: no bacterial colonies at all were detected (zero colonies).

1 to 6 days of age was conducted. A calculated least significant difference (LSD) value of $13.08 \times 10^6$ was obtained at a significance level of $P < 0.05$. Mean values were compared using the calculated LSD and showed the significant differences presented in Fig. 1. While *E. coli* did not colonize any baby delivered by cesarean section, neither male nor female, up to 3 days of age, the bacterium had strongly colonized the babies

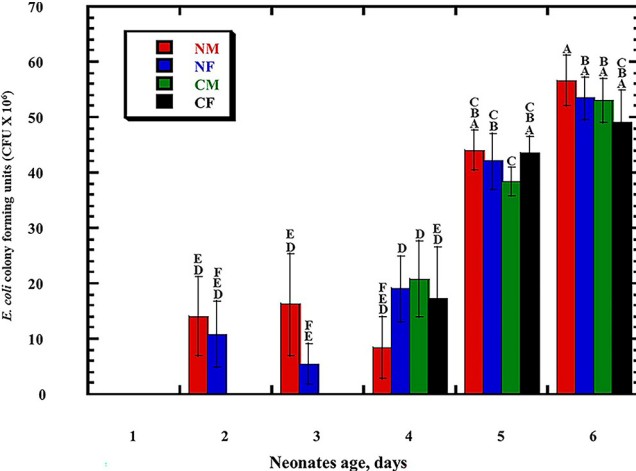

**FIG 1** Determination of *Escherichia coli* CFU (×10⁶) occurrence in first-day meconium and subsequent days' stool of normal (vaginal) birth male neonates (NM), normal birth female neonates (NF), cesarean section surgical delivery male neonates (CM), and cesarean section surgical delivery female neonates (CF). Mean values and standard errors are for results from stool samples of 5 independent babies. Subjects were randomly selected at each age, ranging from 1- to 6-day-old neonates. Bars (mean values) that do not share a letter are significantly different at LSD ($P < 0.05$).

on the fourth day after birth, and the CFU were only slightly lower than for babies delivered vaginally on the fifth and sixth days (Fig. 1). These results indicate a steeper rate of colonization of newborn babies delivered by cesarean section after inoculation from the surrounding environment, which might be attributed to more space for colonization due to late inoculation and, possibly, to the colon having more suitable conditions for colonization at five and six days old. The results presented in this study indicate that colonization of *E. coli* in the colon of newborn babies occurs postnatally, whatever the mode of delivery or the babies' sex.

**Broad postnatal initial colonization of the colon with phylogenetically close *E. coli* strains in healthy newborn babies.** Phylogenetic analysis showed no apparent specific grouping linked to a mode of delivery or sex for all *E. coli* strains isolated from the stool of healthy male and female newborn babies delivered vaginally or by cesarean section (Fig. 6). These results indicate a capability of the various phylogenetically close *E. coli* strains for postnatal initial colonization of the colon, whatever the mode of delivery or sex of the newborn healthy babies, in the first week of their life.

## DISCUSSION

The initial bacterial colonization of neonates' colon is an important and complicated process for the successful establishment of beneficial bacteria in the neonates' colon (4). In this study, the occurrence of *E. coli* was followed in the first-pass meconium and subsequent days' stool samples of healthy newborn babies to explore the prenatal versus postnatal initial colonization of the colon by *E. coli*. No *E. coli* or other enteric bacteria were detected in first-pass meconium on eosin-methylene blue (EMB) agar, LB agar, or MacConkey agar. The first-pass meconium also showed no viable lactic acid bacteria (LAB) on de Man Rososa Sharpe (MRS) agar medium in this study or in a previous study (29) where LAB were detected in second-day stool and later. Despite several reports presenting a microbiome in placenta, amniotic fluid, and meconium, the viability and function of these microbiomes is not fully clear yet. The absence of viable culturable *E. coli* in the first-pass meconium in all subjects investigated and up to the third day in surgically delivered neonates in this study indicates that this bacterium, which is a basic member of the enterobacteria, initiates colonization postnatally and that inoculation of the colon by this bacterium is hard to establish prenatally in uncomplicated pregnancies. The possibility of infection and bactericidal action against bacterial infection in the uterine environment and in the prenatal colon meconium is likely and

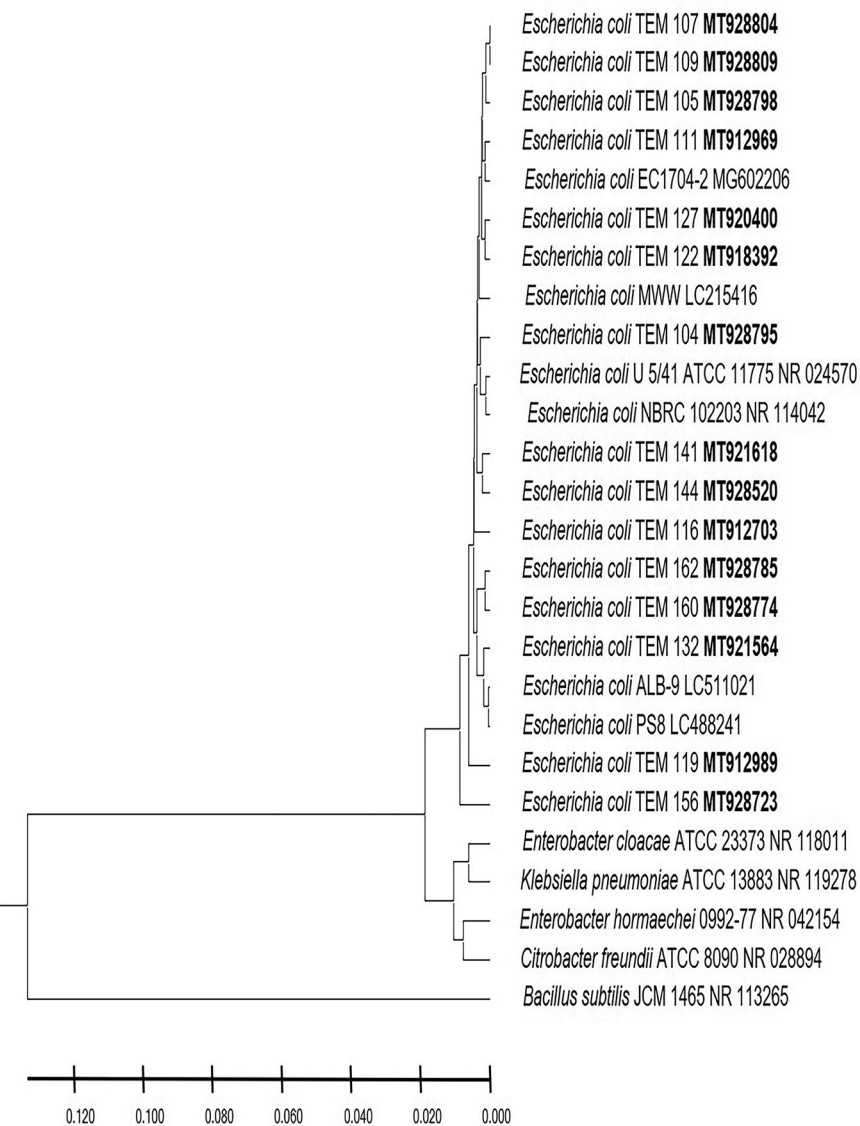

**FIG 2** Phylogenetic analysis of 16S rRNA genes of *Escherichia coli* strains isolated in the first week of life from stool of healthy male neonates born by normal vaginal delivery. A phylogenetic tree of isolated *Escherichia coli* strains shows the relationship with the closest-neighbor bacterial strains from NCBI. The NCBI database accession numbers of *Escherichia coli* strains isolated in this study are shown in boldface. The neighbor-joining tree of isolated *Escherichia coli* strains and other bacterial strains was determined using nearly full-length gene sequences of 16S rRNA and the frequency filter in the MEGA X software analysis package. *Bacillus subtilis* JCM 1465 (accession number NR_113265) was used as an out-group. Scale bar indicates 2% estimated difference in sequence. The strain names and NCBI database accession numbers are shown in the phylogenetic tree and are also listed here: *E. coli* TEM 107, MT928804; *E. coli* TEM 109, MT928809; *E. coli* TEM 105, MT928798; *E. coli* TEM 111, MT912969; *E. coli* EC1704-2, MG602206; *E. coli* TEM 127, MT920400; *E. coli* TEM 122, MT918392; *E. coli* MWW, LC215416; *E. coli* TEM 104, MT928795; *E. coli* U 5/41 ATCC 11775, NR_024570; *E. coli* NBRC 102203, NR_114042; *E. coli* TEM 141, MT921618; *E. coli* TEM 144, MT928520; *E. coli* TEM 116, MT912703; *E. coli* TEM 162, MT928785; *E. coli* TEM 160, MT928774; *E. coli* TEM 132, MT921564; *E. coli* ALB-9, LC511021; *E. coli* PS8, LC488241; *E. coli* TEM 119, MT912989; *E. coli* TEM 156, MT928723; *Enterobacter cloacae* ATCC 23373, NR_118011; *Klebsiella pneumoniae* ATCC 13883, NR_119278; *Enterobacter hormaechei* 0992-77, NR_042154; *Citrobacter freundii* ATCC 8090, NR_028894; *Bacillus subtilis* JCM 1465, NR_113265.

would leave dead or static bacterial cells, and hence, bacterial DNA can be detected, although these microbiomes might be nonviable ones. It was concluded that fetal development proceeds in the absence of an amniotic fluid microbiota in uncomplicated pregnancies, and neonates' microbial colonization was thus predicted to begin after uterine contractions and rupture of the amniotic membrane during delivery (30).

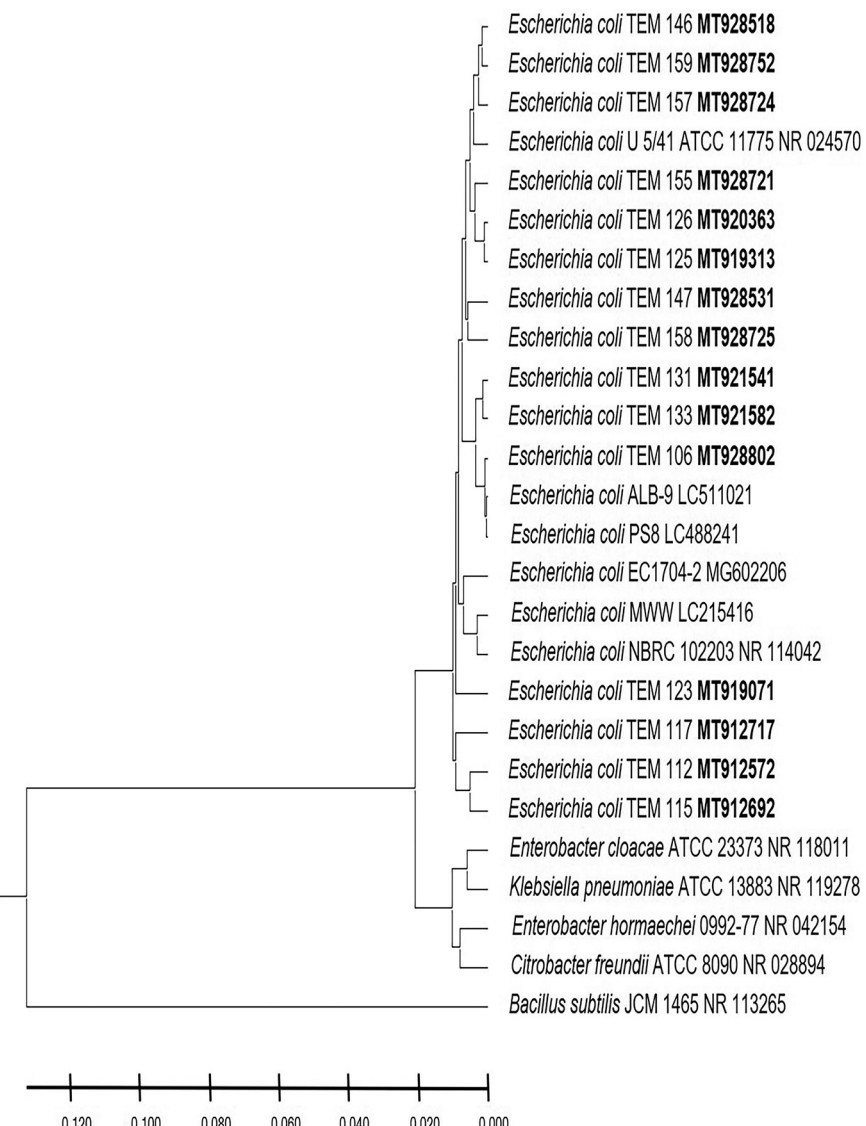

**FIG 3** Phylogenetic analysis of 16S rRNA genes of *Escherichia coli* strains isolated in the first week of life from stool of healthy female neonates born by normal vaginal delivery. A phylogenetic tree of isolated *Escherichia coli* strains shows the relationship with the closest-neighbor bacterial strains from NCBI. The accession numbers of isolated *Escherichia coli* strains are shown in boldface. The neighbor-joining tree of isolated *Escherichia coli* strains and other bacterial strains was determined using nearly full-length gene sequences of 16S rRNA and the frequency filter in the MEGA X software analysis package. *Bacillus subtilis* JCM 1465 (accession number NR_113265) was used as an out-group. Scale bar indicates 2% estimated difference in sequence. The strain names and NCBI database accession numbers are shown in the phylogenetic tree and are also listed here: *E. coli* TEM 146, MT928518; *E. coli* TEM 159, MT928752; *E. coli* TEM 157, MT928724; *E. coli* U 5/41 ATCC 11775, NR_024570; *E. coli* TEM 155, MT928721; *E. coli* TEM 126, MT920363; *E. coli* TEM 125, MT919313; *E. coli* TEM 147, MT928531; *E. coli* TEM 158, MT928725; *E. coli* TEM 131, MT921541; *E. coli* TEM 133, MT921582; *E. coli* TEM 106, MT928802; *E. coli* ALB-9, LC511021; *E. coli* PS8, LC488241; *E. coli* EC1704-2, MG602206; *E. coli* MWW, LC215416; *E. coli* NBRC 102203, NR_114042; *E. coli* TEM 123, MT919071; *E. coli* TEM 117, MT912717; *E. coli* TEM 112, MT912572; *E. coli* TEM 115, MT912692; *Enterobacter cloacae* ATCC 23373, NR_118011; *Klebsiella pneumoniae* ATCC 13883, NR_119278; *Enterobacter hormaechei* 0992-77, NR_042154; *Citrobacter freundii* ATCC 8090, NR_028894; *Bacillus subtilis* JCM 1465, NR_113265.

The human intestine is sterile at birth, and the colonic function of newborn babies is immature (31). The early colonization of vaginally delivered newborn babies' colon by *E. coli* supports that inoculation by this bacterium, in addition to happening via the surrounding environment, occurs through the microbiota in the vaginal canal, which is known to retain *E. coli* (32), during the vaginal delivery process. Inoculation of the

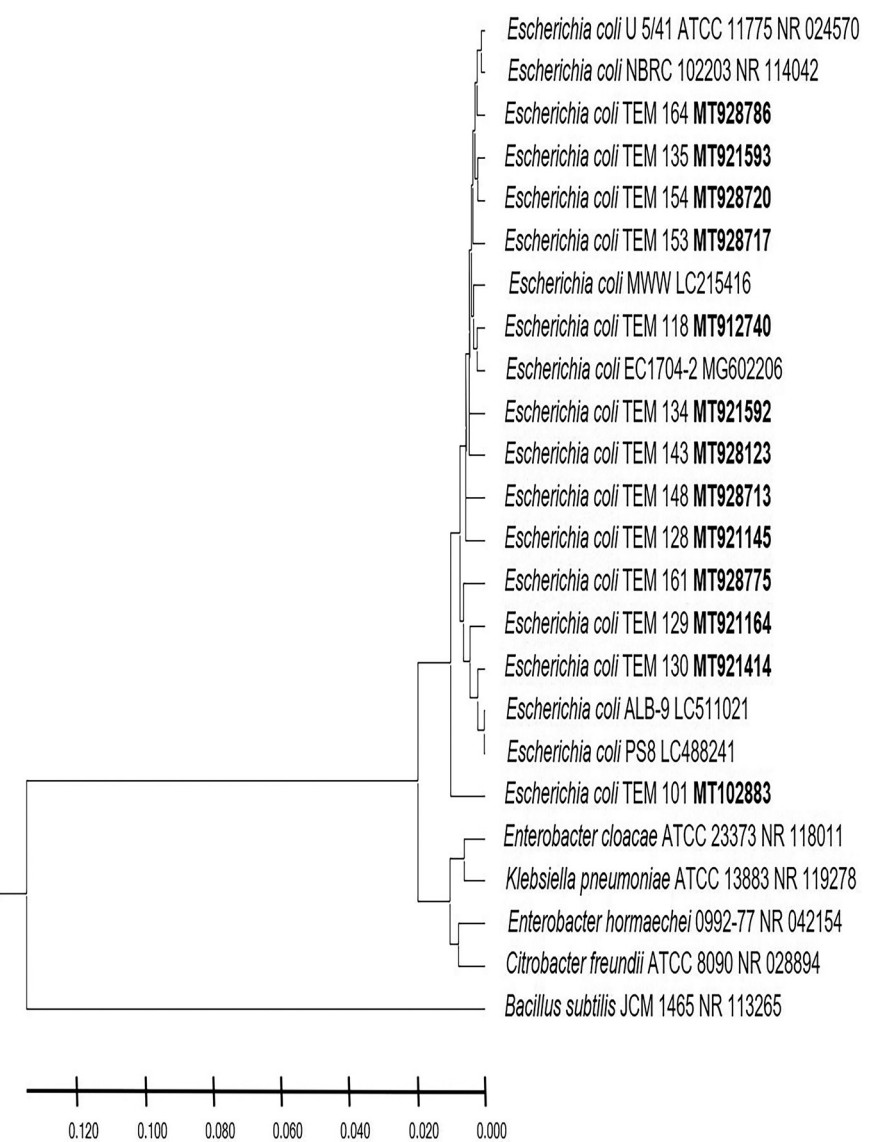

**FIG 4** Phylogenetic analysis of 16S rRNA genes of *Escherichia coli* strains isolated in the first week of life from stool of healthy male neonates delivered by cesarean section. A phylogenetic tree of isolated *Escherichia coli* strains shows the relationship with the closest bacterial neighbor strains from NCBI. The accession numbers of isolated *Escherichia coli* strains are shown in boldface. The neighbor-joining tree of isolated *Escherichia coli* strains and other bacterial strains was determined using nearly full-length gene sequences of 16S rRNA and the frequency filter in the MEGA X software analysis package. *Bacillus subtilis* JCM 1465 (accession number NR_113265) was used as an out-group. Scale bar indicates 2% estimated difference in sequence. The strain names and NCBI database accession numbers are shown in the phylogenetic tree and are also listed here: *E. coli* U 5/41 ATCC 11775, NR _024570; *E. coli* NBRC 102203, NR_114042; *E. coli* TEM 164, MT928786; *E. coli* TEM 135, MT921593; *E. coli* TEM 154, MT928720; *E. coli* TEM 153, MT928717; *E. coli* MWW, LC215416; *E. coli* TEM 118, MT912740; *E. coli* EC1704-2, MG602206; *E. coli* TEM 134, MT921592; *E. coli* TEM 143, MT928123; *E. coli* TEM 148, MT928713; *E. coli* TEM 128, MT921145; *E. coli* TEM 161, MT928775; *E. coli* TEM 129, MT921164; *E. coli* TEM 130, MT921414; *E. coli* ALB-9, LC511021; *E. coli* PS8, LC488241; *E. coli* TEM 101, MT102883; *Enterobacter cloacae* ATCC 23373, NR_118011; *Klebsiella pneumoniae* ATCC 13883, NR _119278; *Enterobacter hormaechei* 0992-77, NR_042154; *Citrobacter freundii* ATCC 8090, NR_028894; *Bacillus subtilis* JCM 1465, NR_113265.

babies' gut by bacteria can thus occur during the babies' passage through the vaginal canal and/or from the surrounding environment just after release, possibly from residual contamination with maternal fecal matter, since most women defecate a little during vaginal birth. It was reported that mothers showing different vaginal microbial communities shared different microorganisms with their newborns, where this might

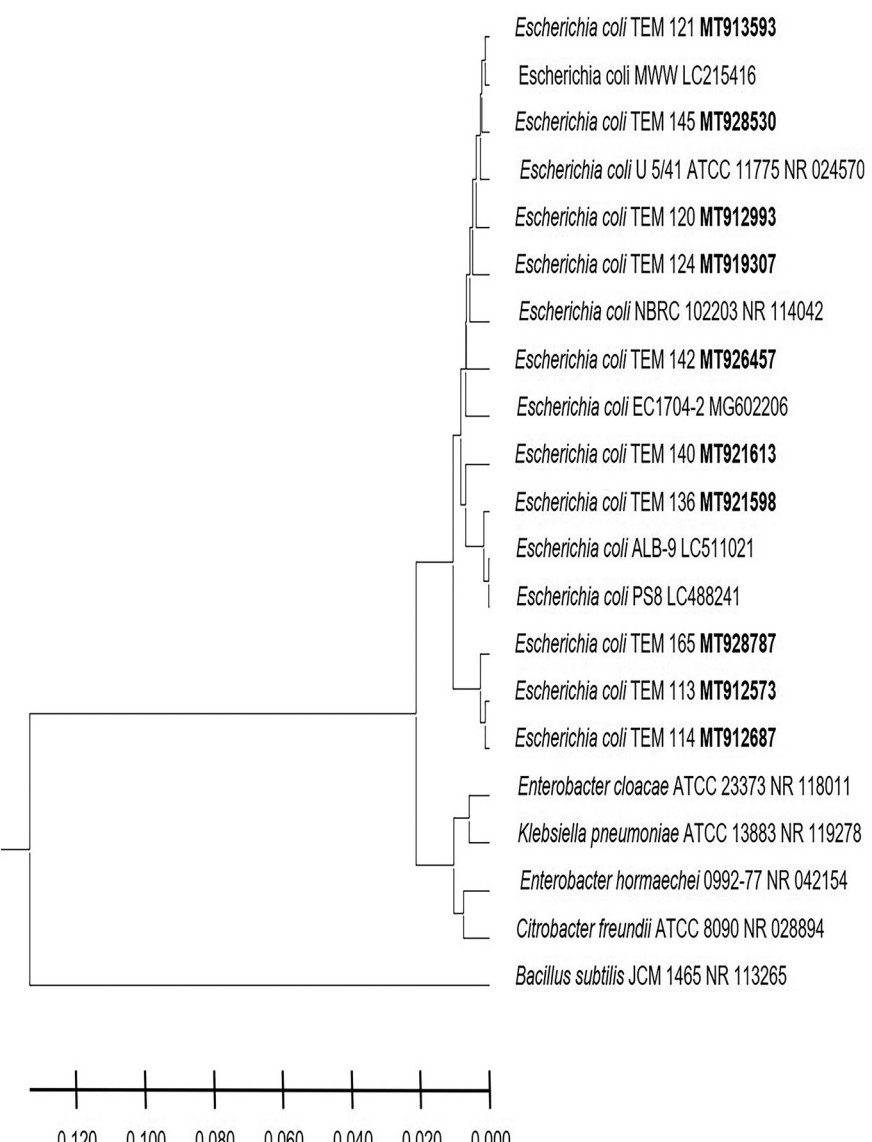

**FIG 5** Phylogenetic analysis of 16S rRNA genes of *Escherichia coli* strains isolated in the first week of life from stool of healthy female neonates delivered by cesarean section. A phylogenetic tree of isolated *Escherichia coli* strains shows the relationship with the closest bacterial neighbor strains from NCBI. The accession numbers of isolated *Escherichia coli* strains are shown in boldface. The neighbor-joining tree of isolated *Escherichia coli* strains and other bacterial strains was determined using nearly full-length gene sequences of 16S rRNA and the frequency filter in the MEGA X software analysis package. *Bacillus subtilis* JCM 1465 (accession number NR_113265) was used as an out-group. Scale bar indicates 2% estimated difference in sequence. The strain names and NCBI database accession numbers are shown in the phylogenetic tree and are also listed here: *E. coli* TEM 121, MT913593; *E. coli* MWW, LC215416; *E. coli* TEM 145, MT928530; *E. coli* U 5/41 ATCC 11775, NR_024570; *E. coli* TEM 120, MT912993; *E. coli* TEM 124, MT919307; *E. coli* NBRC 102203, NR_114042; *E. coli* TEM 142, MT926457; *E. coli* EC1704-2, MG602206; *E. coli* TEM 140, MT921613; *E. coli* TEM 136, MT921598; *E. coli* ALB-9, LC511021; *E. coli* PS8, LC488241; *E. coli* TEM 165, MT928787; *E. coli* TEM 113, MT912573; *E. coli* TEM 114, MT912687; *Enterobacter cloacae* ATCC 23373, NR_118011; *Klebsiella pneumoniae* ATCC 13883, NR_119278; *Enterobacter hormaechei* 0992-77, NR_042154; *Citrobacter freundii* ATCC 8090, NR_028894; *Bacillus subtilis* JCM 1465, NR_113265.

reflect on the initial microbial colonizers of the newborn babies' gut (33). The healthy vaginal microbiota plays an important role in a newborn's health outcome, as the microbial community of the infant's gut is shaped during birth, through the delivery process, which influences the initial assembly of microbiota in the gut (34, 35). All newborn babies that were subjects in this study were chosen from those given normal

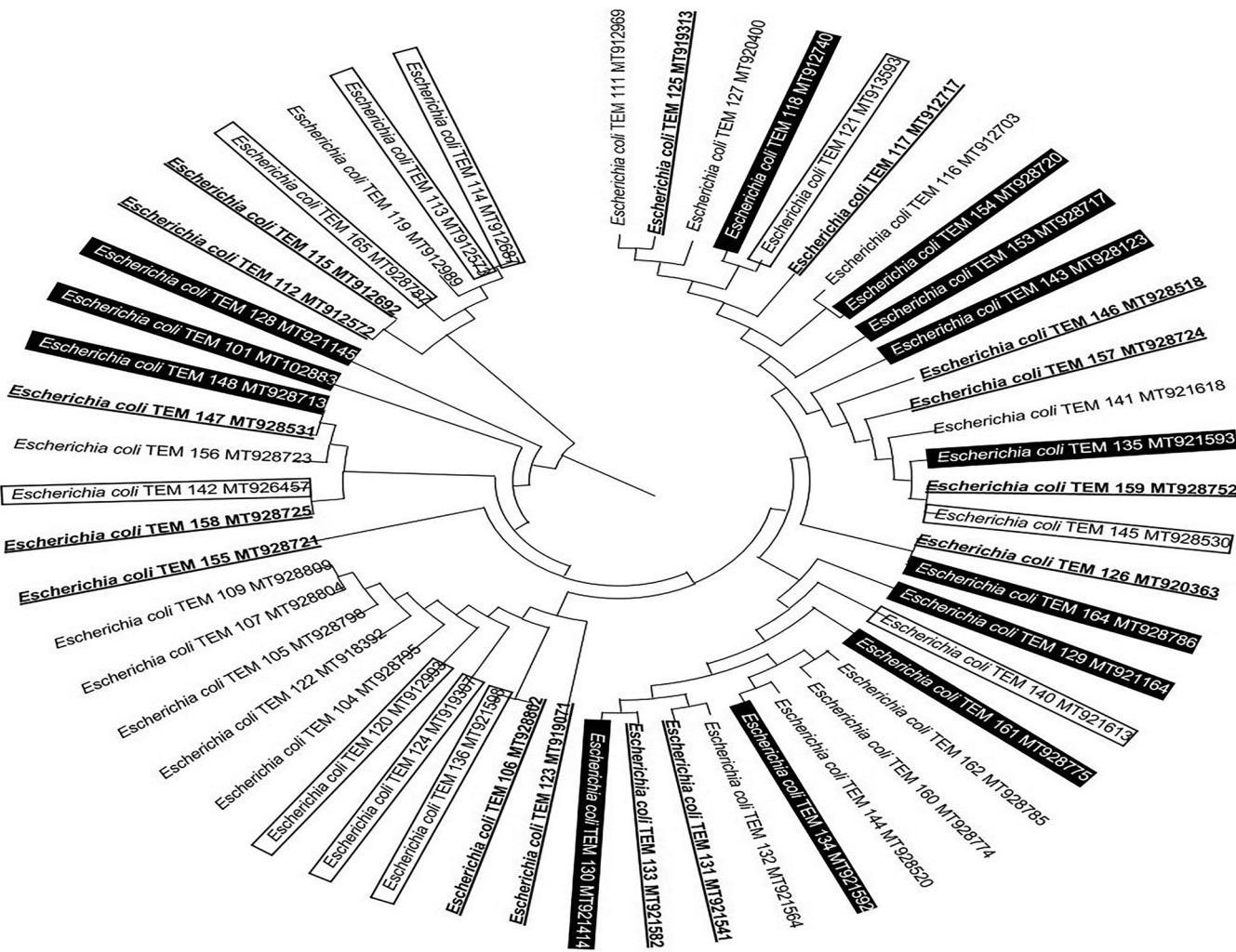

**FIG 6** Phylogenetic relationship topology tree of all *Escherichia coli* strains isolated in the first week of life from stool of healthy male and female newborn babies born vaginally or by cesarean section. The NCBI database accession number of each strain is shown. The isolated *Escherichia coli* strains from neonates born by each type of delivery and of each sex are shown as unmarked, underlined boldface, inside a text box, and white in a black background for strains isolated from healthy male and female babies born vaginally or by cesarean section, respectively. The topology tree of all isolated *Escherichia coli* strains was determined using nearly full-length gene sequences of 16S rRNA and the frequency filter in the MEGA X software analysis package. The strain names and NCBI database accession numbers are shown in the phylogenetic tree and are also listed here: *E. coli* TEM 111, MT912969; *E. coli* TEM 125, MT919313; *E. coli* TEM 127, MT920400; *E. coli* TEM 118, MT912740; *E. coli* TEM 121, MT913593; *E. coli* TEM 117, MT912717; *E. coli* TEM 116, MT912703; *E. coli* TEM 154, MT928720; *E. coli* TEM 153, MT928717; *E. coli* TEM 143, MT928123; *E. coli* TEM 146, MT928518; *E. coli* TEM 157, MT928724; *E. coli* TEM 141, MT921618; *E. coli* TEM 135, MT921593; *E. coli* TEM 159, MT928752; *E. coli* TEM 145, MT928530; *E. coli* TEM 126, MT920363; *E. coli* TEM 164, MT928786; *E. coli* TEM 129, MT921164; *E. coli* TEM 140, MT921613; *E. coli* TEM 161, MT928775; *E. coli* TEM 162, MT928785; *E. coli* TEM 160, MT928774; *E. coli* TEM 144, MT928520; *E. coli* TEM 134, MT921592; *E. coli* TEM 132, MT921564; *E. coli* TEM 131, MT921541; *E. coli* TEM 133, MT921582; *E. coli* TEM 130, MT921414; *E. coli* TEM 123, MT919071; *E. coli* TEM 106, MT928802; *E. coli* TEM 136, MT921598; *E. coli* TEM 124, MT919307; *E. coli* TEM 120, MT912993; *E. coli* TEM 104, MT928795; *E. coli* TEM 122, MT918392; *E. coli* TEM 105, MT928798; *E. coli* TEM 107, MT928804; *E. coli* TEM 109, MT928809; *E. coli* TEM 155, MT928721; *E. coli* TEM 158, MT928725; *E. coli* TEM 142, MT926457; *E. coli* TEM 156, MT928723; *E. coli* TEM 147, MT928531; *E. coli* TEM 148, MT928713; *E. coli* TEM 101, MT102883; *E. coli* TEM 128, MT921145; *E. coli* TEM 112, MT912572; *E. coli* TEM 115, MT912692; *E. coli* TEM 165, MT928787; *E. coli* TEM 119, MT912989; *E. coli* TEM 113, MT912573; *E. coli* TEM 114, MT912687.

breast feeding. Future studies comparing the effects of breast and formula feeding on the postnatal initial microbiome colonizing the neonates' colon would be of interest. Several reports attributed the bacterial inoculation of surgically delivered newborn babies to the feeding process (26). This study showed that the initial colonization by *E. coli* strains in the neonates' colon seemed to be fundamentally occurring postnatally, in the first week of their life, whatever the mode of delivery or sex of the newborn healthy babies. In spite of the early postnatal colonization of newborn babies' colon by *E. coli* detected in this study, a delay in colonization was detected in other studies, where only 61% of the infants were positive by 2 months of age, and many factors might affect the colonization (36). Despite concern about the influence exerted on

neonates' meconium microbiota by antibiotic exposure of healthy mothers prior to labor or to the operative period for cesarean section, the results in this study showed no *E. coli* colonization in the first-pass meconium of healthy newborn babies delivered vaginally, confirming a postnatal colonization. Future studies would be of interest to investigate the effect of antibiotic exposure of mothers requiring health care prior to vaginal or cesarean section delivery on the neonates' postnatal bacterial colonization of the colon.

One of the possible roles of the early postnatal colonization of *E. coli* as a basic member of the bacteria initially colonizing the colon of newborn babies is its expected function of scavenging molecular oxygen gas. This helps in establishing and maintaining the micro-aerobic and anaerobic conditions that are mandatory for colonization of the colon by some microaerobic and anaerobic beneficial bacteria widely detected in the colon of healthy babies. Thus, *E. coli* might, in addition to other facultative anaerobes, act as an initial servant bacterium that consumes molecular oxygen gas, establishing anaerobiosis for colon colonization by microaerobic and anaerobic bacteria like *Lactobacilli*. The *Lactobacilli* are members of the initial colonizing bacteria of the colon in healthy newborn babies (29). *Lactobacillus* spp. are usually found in babies' colon, where they play important roles in health (37, 38). Other beneficial microaerobes and anaerobes in neonates' colon would also require the microaerobic and anaerobic conditions that can be established by the facultative anaerobic bacteria, including *E. coli*, which consume oxygen and exist permanently in the microbiota for such functions as maintaining the microaerobic and anaerobic conditions required for viability of the beneficial microaerobic and anaerobic bacteria. The consumption of oxygen by aerobic and facultative anaerobic bacteria, along with other $O_2$ consumption mechanisms in the human gut, was suggested to maintain the gut lumen in a deeply anaerobic status that is an important condition for obligate anaerobes (39). Understanding the interactions among the microbiota, the host, and pathogenic bacteria can help in exploring strategies to manipulate the gut microbiota against enteric pathogens (40). Interestingly, the commensal *Enterobacteriaceae* species *E. coli* was found to protect against intestinal colonization of neonatal chickens by pathogenic *Salmonella* through their competition for oxygen consumption, which reduces the expansion of this pathogen that can occur under aerobic respiration conditions (41). The vitamins known to be produced by *E. coli*, such as vitamin $K_2$ (menaquinone) (42, 43) and vitamin $B_1$ (44, 45), might be important at this early stage starting from the first week of the babies' life, possibly for healthy body development. In spite of the pathogenicity of some *E. coli* strains (46, 47), this bacterium is a basic member of the human colon microbial community (18, 48), where many strains other than pathogenic ones were reported as commensal and some as beneficial to health (49–52). Since all the babies selected for this study were in a healthy state, the early colonization by *E. coli* strains identified in this study indicates a commensal status of these strains of the bacterium, whose postnatal colonization of the colon possibly helps to inhibit colonization by pathogenic strains of the bacterium or other bacterial taxa. Some commensal and health-beneficial strains of *E. coli* were reported to inhibit pathogenic *E. coli* strains and other taxa of pathogenic bacteria (53–57). Although the small intestine of healthy babies actively produces lactase, for hydrolyzing the milk sugar lactose to allow absorption of the monosaccharides glucose and galactose that are produced, some amount of lactose might pass in an undigested form to the babies' colon, where it can be utilized by lactic acid- and other lactose-fermenting bacteria, such as *E. coli*, for a different net product. This ability of *E. coli* to utilize lactose might help in its early colonization of the neonates' colon. Thus, along with the early colonization of the neonates' colon by lactic acid bacteria (LAB) (29), *E. coli* also has suitable nutritional conditions, explaining its good capacity for initial colonization of the neonates' colon, where milk containing lactose is the only diet provided to the neonates. This also supports our conclusion that the initial colonization of *E. coli* is postnatal, when milk feeding of newborn babies confers a viable nutritional requirement of this bacterium and many other health-beneficial members of lactose-fermenting bacterial taxa. The suitable nutritional conditions for these bacteria would help in fulfilling their health-beneficial roles in this early stage of the neonates'

body development. However, a defect in lactase production in babies' or adults' small intestine would lead to an enhanced amount of lactose and, hence, enhanced fermentation by lactose-fermenting bacteria like *E. coli*, leading to health troubles and risk of lactose intolerance and requiring feeding with lactose-free milk or other treatments. The phylogenetic analysis of the 16S rRNA-encoding genes of the isolated *E. coli* strains showed no apparent specific grouping linked to mode of delivery or sex, indicating a capacity of the various phylogenetically close strains for postnatal colon colonization in the first week of life in all investigated male and female healthy newborn babies born by the two modes of delivery. Future studies using various other molecular-biological tools would be of interest for characterizing the broad postnatal ability of phylogenetically close strains of *E. coli* to establish initial colonization of the colon in male and female newborn babies delivered vaginally or by cesarean section. The absence of *E. coli* in meconium in all neonates investigated indicates possible antimicrobial properties present in it for protecting neonates from infection during the delivery process, whereas *E. coli* colonization occurs after the first-pass meconium by the 1st day in the case of vaginal birth and by the 3rd day in the case of cesarean section surgical delivery. Antimicrobial peptides were previously detected in meconium (58), which might play a substantial role as an antimicrobial protection against prenatal colon colonization. Future studies would be of interest to explore the specific antimicrobial action and whole spectrum and forms of antimicrobial agents in meconium against inoculation of the neonates by various pathogenic and nonpathogenic bacteria in the vaginal microbiome. Other possible factors, including the suitability of nutritional conditions of meconium in the colon for viable prenatal microbial colonization compared to the suitability of postnatal milk feeding nutritional conditions, would be of interest for future studies. Because of their high degree of conservation in closely related species (59, 60) and their several copies within the genome (60–64), it has been shown that sequence analysis of the *gyrB*, *atpD*, *infB*, and *rpoB* genes is a useful alternative to 16S rRNA analysis for the classification of the enterobacteria *Salmonella*, *Shigella*, and *E. coli* (60, 65–67) for exploring the phylogenetic relationships at the species level, particularly for closely related species (67).

Further studies in more places and with more bacterial genera and larger numbers of samples would be of interest for exploring the controlling mechanism of the initial colonization of the colon in healthy neonates with commensal and health-beneficial bacteria. In this study, regular *Taq* polymerase was used. Although high-fidelity polymerase is usually used in amplification for molecular cloning, it also can be preferable for use in future studies for the amplification and sequencing of 16S rRNA for phylogenetic analysis. All neonates in this study were healthy and received no special health care. It would be of interest in future studies to investigate the effects of various health care conditions and antimicrobial medications on postnatal initial bacterial colonization of the colon in neonates receiving health care, such as neonates of preterm birth in comparison to healthy newborn babies.

In conclusion, the results of this study suggest a naturally controlled postnatal initial colonization of neonates' colon by commensal beneficial bacteria, where the prenatal colon's meconium might be a part of the mechanisms controlling the initial colonization of the babies' gut for protecting against infection during delivery.

## MATERIALS AND METHODS

The occurrence of *E. coli* was investigated in first-day meconium and subsequent days' stool of healthy neonates in Medina, Kingdom of Saudi Arabia (KSA). The studies involving human participants were reviewed and approved by the Institutional Review Board (IRB) in Medina (IRB Committee head, Abduhameed Alsubhi). Informed consent was obtained from parents for the collection of infant stools.

**Initial colonization of colon by *E. coli* in healthy neonates in the first week of life.** The initial colonization of the colon by *E. coli* in healthy neonates in the first week of life was followed by investigating its occurrence in meconium and in stool of healthy newborn babies on the subsequent 5 days. All neonates were selected as being healthy. This was the basic condition for selecting neonates in this study. Different neonates were chosen to have a wide view of the initial colonization. Different healthy neonates at 1 day of age were selected randomly for obtaining the first-pass meconium, while subsequent days' stool samples were obtained from babies 2 to 6 days old. A total of 120 newborn babies were investigated (20 neonates of each age from 1 to 6 days; for each day of age, 10 were born vaginally and 10 were delivered surgically, and equal numbers of male and female neonates were chosen). Once first-

pass meconium collected on the first day after birth or subsequent days' stool samples were obtained normally in diapers, they were sent to the bacteriology laboratory and used directly for examining the occurrence of *E. coli* and determining its CFU. The occurrence of *E. coli* in meconium and subsequent days' stool samples was conducted using eosin-methylene blue (EMB) selective agar medium. CFU were determined by most probable number (MPN) techniques (68). The colonies of *E. coli*, showing a metallic green sheen, were followed on the selective EMB agar, where the CFU of the bacterium was estimated from the lowest number of colonies on the EMB agar plate in serial dilutions of the inoculum, followed by morphological and biochemical characterizations as outlined in Bergey's Manual (69). From the bacterial colonies from the stool sample of each infant that were identified by morphological and biochemical characterizations, a single bacterial isolate colony per infant was selected from each of the healthy babies for further identification by phylogenetic analysis of the 16S rRNA-encoding gene sequence.

**PCR amplification of the 16S rRNA gene.** The Promega Wizard genomic DNA purification kit (Promega Corporation, Madison, WI, USA) was used to extract the genomic DNA of *E. coli* bacterial cultures according to the manufacturer's instructions. The universal forward (27F, 5′-AGAGTTTGATC[A/C]TGGCTCAG-3′) and reverse (1492R, 5′-G[C/T]TACCTTGTTACGACTT-3′) primers (70) were used to amplify a nearly full-length sequence of the 16S rRNA-encoding gene by PCR, using a reaction mixture (25 $\mu$l) composed of 2.5 $\mu$l 10× *Taq* buffer (100 mM Tris-HCl, pH 8), 1.25 mM $MgCl_2$, 100 $\mu$M deoxynucleoside triphosphates (dNTPs) (Invitrogen, Carlsbad, CA, USA), 1.2 $\mu$M forward and reverse primers, 0.5 U *Taq* DNA polymerase (Invitrogen, USA), and about 5 ng template genomic DNA. A thermal cycler (model 2720; Applied Biosystems, Foster City, CA, USA) was used for the PCR amplification, which was performed with the following PCR program: 95°C for 5 min (initial denaturation) and then 35 amplification cycles of 94°C for 1 min (denaturation), 56°C for 1 min (annealing), and 72°C for 1 min (extension), followed by a final extension at 72°C for 10 min. Agarose gel electrophoresis was used for analyzing the PCR amplification products on agarose (1%) gels containing ethidium bromide (5 $\mu$g/ml) with DNA size marker (1 kb Plus DNA ladder; Invitrogen, USA).

**Nucleotide sequence analysis.** The PCR products were purified and cycle sequenced at the Macrogen Korea sequencing facility, (Seoul, South Korea). Direct cycle sequencing of the purified PCR product was conducted using the same forward and reverse primers in both directions and the automated florescent dye terminator sequencing method (71) at the Macrogen Korea sequencing facility, (Seoul, South Korea) in the 3730XL DNA analyzer (Applied Biosystems, CA, USA). The sequence reads were assembled and compared with the closest matches in GenBank by using the nucleotide-nucleotide BLAST search tool of the National Center for Biotechnology and Information (NCBI) server at www.ncbi .nlm.nih.gov/blast/Blast.cgi. Alignments of 16S rRNA gene sequences were performed by using Clustal W1.83 XP (72). The derived phylogenetic tree of the 16S rRNA gene sequences was constructed using the neighbor-joining method (73) by MEGA X software (74). An outgroup (*Bacillus subtilis* strain JCM 1465, accession number NR_113265) was used.

**Statistical analysis.** All of the healthy neonates who were subjects in the study were randomly selected. The results for *E. coli* CFU in male and female newborn babies born vaginally or by cesarean section were subjected to one-way analysis of variance (ANOVA) using Microsoft Excel, and the least significant difference (LSD) value was calculated. Mean values were compared using the calculated LSD value at a significance level of $P < 0.05$.

**Data availability.** The partial nucleotide sequences of the 16S rRNA-encoding genes of *E. coli* strains isolated from stool samples obtained in the first week of life from healthy male and female newborn babies delivered vaginally or by cesarean section were deposited in NCBI's GenBank nucleotide sequence database under accession numbers outlined in Tables S1 to S4. The data sets generated for this study can be found in the online repositories. The names of the repository/repositories and accession number(s) can be found in the article/supplemental material.

## SUPPLEMENTAL MATERIAL

Supplemental material is available online only.

**SUPPLEMENTAL FILE 1**, PDF file, 0.1 MB.

## ACKNOWLEDGMENTS

All authors listed made a substantial, direct, and intellectual contribution to the work and approved it for publication. The authors contributed equally to this work.

We declare that the research was conducted in the absence of any commercial or financial relationships that could be construed as a potential conflict of interest.

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
