## [Reviewer comments · Microbiology Spectrum]

Microbiology Spectrum

Prenatal versus postnatal initial colonization of healthy neonates' colon ecosystem by the enterobacterium *Escherichia coli*

Mohammad Al-Balawi and Fatthy Morsy

Corresponding Author(s): Fatthy Morsy, Taibah University

Review Timeline:

Submission Date:	May 24, 2021
Editorial Decision:	July 18, 2021
Revision Received:	August 20, 2021
Editorial Decision:	September 19, 2021
Revision Received:	September 22, 2021
Accepted:	September 23, 2021

Editor: Laxmi Yeruva

Reviewer(s): The reviewers have opted to remain anonymous.

Transaction Report:

DOI: <https://doi.org/10.1128/Spectrum.00379-21>

July 18, 2021

Prof. Fatthy M Morsy
Taibah University
Biology
Elsalam
Al-Madinah, Al-Madinah
Saudi Arabia

Re: Spectrum00379-21 (Prenatal versus postnatal initial colonization of healthy neonates' colon ecosystem by the enterobacterium Escherichia coli)

Dear Prof. Fatthy M Morsy:

Both reviewers have raised excellent points about the quality of presentation, modifying the discussion/conclusions to fit with the data presented and not extrapolating the observations. Additionally, manuscript needs editing/requires careful proofing.

Thank you for submitting your manuscript to Microbiology Spectrum. When submitting the revised version of your paper, please provide (1) point-by-point responses to the issues raised by the reviewers as file type "Response to Reviewers," not in your cover letter, and (2) a PDF file that indicates the changes from the original submission (by highlighting or underlining the changes) as file type "Marked Up Manuscript - For Review Only". Please use this link to submit your revised manuscript - we strongly recommend that you submit your paper within the next 60 days or reach out to me. Detailed information on submitting your revised paper are below.

Link Not Available

Sincerely,

Laxmi Yeruva

Journals Department
Reviewer comments:

Reviewer #1 (Comments for the Author):

The study reported the presence of viable culturable Escherichia coli in meconium and later fecal samples. Traditional culture methods and specific sanger sequencing were used to identify E.coli strains in healthy neonatal gut. Results showed that E.coli was not presence on meconium samples and C-section infant had a delay in E.coli detection compared to vaginal neonates.

major comments:

- Meconium sample is a sample that is collected <48h after birth, however, in terms of microbiota exposition, this matters. After 24h, the neonate is exposed to maternal skin, milk and oral microbes and also, family and environmental bacteria. Please, more detailed information is needed on sampling time.
- Did the authors check other culture media as LB and/or MacConkey?
- Please, could you clarify if your method had a specific detection limit? there is a big difference between non detected and

detected samples. The range is Non-detected and when the bacteria is present, the amount is approx 1000000 CFU/gr. Could you detail the process and the technique? Does 10^6 the detection threshold? if yes, it is too high considering that meconium is a low biomass sample.

- Please, include details and discussion on the problems related to the 16S fragment sequencing and the differentiation between E. coli/Salmonella/Shigella and also, the potential use of other genes as gyrB, atpD, infB, rpoB among others.
- Figure 1 is not clear. Microbes had exponential growth and if data would be reported in log manner, the samples are more or less in same logarithmic unit (log 6)
- please include a paragraph on limitations of the study.
- conclusion needs to be focused to the data obtained in the study.

minor comments:

- please, change "microflora" by "microbiota"
- please, include the exact days where fecal samples were analyzed. The "subsequent days" is not sufficient information.

Reviewer #2 (Comments for the Author):

This manuscript reports culture and 16S sequencing data for stool samples collected from human infants. The bacterial characterization is limited to E. Coli. The sample size is small. The work is of interest.

However, the final two bulleted "Highlights" in the supplemental file are not conclusions that can be drawn on the basis of the data presented in this manuscript.

It is not entirely clear, but it seems that the study design is cross-sectional. Is it correct that different infants are sampled at each time point presented.

More information about the demographics, diet (formula versus human milk), and antibiotic use for the participating infants needs to be provided to give the appropriate context for interpretation of the research results.

What is the standard of care for c-section deliveries at the hospital? Which antibiotics are used? How long could they be expected to be in the bloodstream of the mother and infant? Could that regimen affect the viability of microbes in the infant?

Detailed information about sample collection is required in the methods section. Were the samples collected from diapers? Or, was sampling done using swabs? By whom and when were the samples collected? Were samples stored at all before processing? If so, how?

Was regular Taq (line 135) not high fidelity polymerase used for preparation of the 16S genes for sequencing? If so, this could introduce errors in the sequences and should be listed as a limitation. This is especially true in light of the length of PCR product generated.

With respect to the collection of colonies for determination of E. Coli strains, please clarify if a single colony per infant was selected or how many colonies were sampled per infant for the sequencing.

The discussion section (and results section) could be reduced in length by removing the repeated statement of the same results that can found in many, many locations throughout the manuscript.

If you have not cited the work of Andreas Baumler in your discussion of the de-oxygenation of the gut and the role that plays in establishment of the gut microbiota, then please do.

In several places, it is suggested that the vagina might be the source of the E. Coli. However, it is more likely that fecal material would be the source of E. Coli.

Consider using the word "vaginal" instead of "normal" with reference to birth.

Consider using the word, "sex" instead of the word, "gender."

Staff Comments:

Preparing Revision Guidelines

To submit your modified manuscript, log onto the eJP submission site at <https://spectrum.msubmit.net/cgi-bin/main.plex>. Go to Author Tasks and click the appropriate manuscript title to begin the revision process. The information that you entered when you

first submitted the paper will be displayed. Please update the information as necessary. Here are a few examples of required updates that authors must address:

For complete guidelines on revision requirements, please see the Instructions to Authors at [link to page]. **Submissions of a paper that does not conform to Microbiology Spectrum guidelines will delay acceptance of your manuscript.**

Please return the manuscript within 60 days; if you cannot complete the modification within this time period, please contact me. If you do not wish to modify the manuscript and prefer to submit it to another journal, please notify me of your decision immediately so that the manuscript may be formally withdrawn from consideration by Microbiology Spectrum.

If you would like to submit an image for consideration as the Featured Image for an issue, please contact Spectrum staff.

Point-to-Point Response to Reviewers

Manuscript ID: Spectrum00379-21

The authors are grateful to the precious suggestions, constructive comments and careful corrections made by the two anonymous reviewers for further improvements of this paper. The manuscript has been revised according to the reviewers' comments.

Response to Reviewer #1

Reviewer comments:

Reviewer #1 (Comments for the Author):

The study reported the presence of viable culturable *Escherichia coli* in meconium and later fecal samples. Traditional culture methods and specific sanger sequencing were used to identify *E. coli* strains in healthy neonatal gut. Results showed that *E. coli* was not presence on meconium samples and C-section infant had a delay in *E. coli* detection compared to vaginal neonates.

major comments:

- Meconium sample is a sample that is collected <48h after birth, however, in terms of microbiota exposition, this matters. After 24h, the neonate is exposed to maternal skin, milk and oral microbes and also, family and environmental bacteria. Please, more detailed information is needed on sampling time.

Response to the reviewer comment: Done; Once first-pass meconium collected in first day after birth or subsequent days stool samples were obtained normally in diapers, they were sent to the bacteriology laboratory and directly used for examining the occurrence of *E. coli* and determining its CFU (Colony Forming Units). This was outlined in the revised form of the manuscript.

- Did the authors check other culture media as LB and/or MacConkey?

Response to the reviewer comment: Done, No *E. coli* or other enteric bacteria were detected in first-pass meconium on EMB, LB agar or MacConkey agar. The first-pass meconium also showed no viable lactic acid bacteria (LAB) on MRS agar medium in this study or in a previous study (Al-Balawi and Morsy, 2020) where LAB were detected in 2nd day stool and above. This was outlined in the revised form of the manuscript.

- Please, could you clarify if your method had a specific detection limit? there is a big difference between non detected and detected samples. The range is Non-detected and when the bacteria is present, the amount is aprox 1000000 CFU/gr. Could you detail the process and the technique? Does 10⁶ the detection threshold? if yes, it is too high considering that meconium is a low biomass sample.

Response to the reviewer comment: Done; Non detected in this study means no bacterial colonies at all (Zero colonies detected). This was outlined under table 1 in the revised form of the manuscript.

- Please, include details and discussion on the problems related to the 16S fragment sequencing and the differentiation between *E. coli/Salmonella/Shigella* and also, the potential use of other genes as *gyrB*, *atpD*, *infB*, *rpoB* among others.

Response to the reviewer comment: Done; Because of its high degree of conservation in closely related species (Spröer *et al.*, 1999; Paradis *et al.*, 2005) and its several copies within the genome (Paradis *et al.*, 2005; Kembel *et al.*, 2012; Větrovský and Baldrian, 2013; Louca *et al.*, 2018; Ibal *et al.*, 2019), it has been shown that *gyrB*, *atpD*, *infB*, *rpoB* genes sequence analysis is a useful alternative to 16S rRNA analysis for the classification of the enterobacteria *Salmonella*, *Shigella* and *E. coli* (Mollet *et al.*, 1997; Hedegaard *et al.*, 1999; Fukushima *et al.*, 2002; Paradis *et al.*, 2005) for exploring the phylogenetic relationships at the species level in particular of closely related ones (Fukushima *et al.*, 2002). This was outlined in the revised form of the manuscript.

Louca, S., Doebeli, M., and Parfrey, L.W. (2018) Correcting for 16S rRNA gene copy numbers in microbiome surveys remains an unsolved problem. *Microbiome* 6(1), 41.

Kembel, S. W., Wu, M., Eisen, J. A., and Green, J. L. (2012) Incorporating 16S gene copy number information improves estimates of microbial diversity and abundance. *PLOS Comput. Biol.* 8(10), 1-11.

Ibal, J. C., Pham, H. Q., Park, C. E., and Shin, J. H. (2019). Information about variations in multiple copies of bacterial 16S rRNA genes may aid in species identification. *PloS one*, 14(2), e0212090.

Větrovský, T., and Baldrian, P. (2013). The variability of the 16S rRNA gene in bacterial genomes and its consequences for bacterial community analyses. *PloS one*, 8(2), e57923.

Spröer, C., Mendrock, U., Swiderski, J., Lang, E., and Stackebrandt, E. (1999). The phylogenetic position of *Serratia*, *Buttiauxella* and some other genera of the family *Enterobacteriaceae*. *Int. J. Syst. Bacteriol.* 49(Pt 4), 1433-1438

Mollet, C., Drancourt, M. and Raoult, D. (1997). rpoB sequence analysis as a novel basis for bacterial identification. *Mol. Microbiol.* 26, 1005-1011.

Hedegaard, J., Steffensen, S. A., Nørskov-Lauritsen, N., Mortensen, K. K. and Sperling-Petersen, H. U. (1999). Identification of *Enterobacteriaceae* by partial sequencing of the gene encoding translation initiation factor 2. *Int. J. Syst. Bacteriol.* 49, 1531-1538

Fukushima, M., Kakinuma, K., and Kawaguchi, R. (2002). Phylogenetic analysis of *Salmonella*, *Shigella*, and *Escherichia coli* strains on the basis of the gyrB gene sequence. *J. Clin. Microbiol.* 40(8), 2779-2785.

Paradis, S., Boissinot, M., Paquette, N., Bélanger, S. D., Martel, E. A., Boudreau, D. K., Picard, F. J., Ouellette, M., Roy, P. H., and Bergeron, M. G. (2005). Phylogeny of the Enterobacteriaceae based on genes encoding elongation factor Tu and F-ATPase beta-subunit. *Int. J. Syst. Evol. Microbiol.* 55(Pt 5), 2013-2025.

-Figure 1 is not clear. Microbes had exponential growth and if data would be reported in log manner, the samples are more or less in same logarithmic unit (log 6)

Response to the reviewer comment: Done; As the data for colonization are from different babies we avoided our comparison in a growth phases manner. The following revised text was outlined in the revised form of the manuscript: (While *E. coli* colonized no baby of cesarean section surgical delivery neither male nor female up to babies of 3 days old, the bacterium sharply colonized the babies at 4th day after

birth and the CFU was slightly lower than that of normal birth at 5th and 6th day (Fig.1). These results indicate a sharper colonization after inoculation from the surrounding environment of newborn babies of cesarean section surgical delivery which might be attributed to a more space for colonization due to late inoculation and to possibly a more suitability of colonization condition at this age of 5 and 6 days old.)

- please include a paragraph on limitations of the study.

Response to the reviewer comment: Done; the following paragraph was outlined in the revised form of the manuscript.

Further future studies on more places and with more bacterial genera and number of samples would be of interest for exploring the controlling mechanism of the initial colonization of colon in healthy neonates with commensal and health beneficial bacteria. In this study, regular *Taq* polymerase was used. Although high fidelity polymerases, is usually used in amplification for molecular cloning, it also can be preferably used in future studies in amplification and sequencing of 16S rRNA for phylogenetic analysis. All neonates' subjects in this study were chosen healthy and received no special health care. It would be of interest in future studies to investigate the effect of various health care conditions and antimicrobial medications on postnatal initial bacterial colonization of colon in health care subjected neonates such as neonates of preterm birth in comparison to healthy newborn babies.

- conclusion needs to be focused to the data obtained in the study.

Response to the reviewer comment: Done, revised.

minor comments:

-please, change "microflora" by "microbiota"

Response to the reviewer comment: Done

- please, include the exact days where fecal samples were analyzed. The "subsequent days" is not sufficient information.

Response to the reviewer comment: Done; As mentioned above; Once first-pass meconium collected in first day after birth or subsequent days stool samples were obtained normally in diapers, they were sent to the bacteriology laboratory and

directly used for examining the occurrence of *E. coli* and determining its CFU. This was outlined in the revised form of the manuscript.

Response to Reviewer #2

Reviewer #2 (Comments for the Author):

This manuscript reports culture and 16S sequencing data for stool samples collected from human infants. The bacterial characterization is limited to *E. coli*. The sample size is small. The work is of interest.

However, the final two bulleted "Highlights" in the supplemental file are not conclusions that can be drawn on the basis of the data presented in this manuscript.

Response to the reviewer comment: Done; Highlights were revised.

It is not entirely clear, but it seems that the study design is cross-sectional. Is it correct that different infants are sampled at each time point presented.

Response to the reviewer comment: Done; Different neonates were chosen to have wide view of the initial colonization where healthy different neonates were selected randomly of age one day for obtaining the first-pass meconium while subsequent days stool samples were obtained from babies of age 2 to 6 days old. This was outlined in the revised form of the manuscript.

More information about the demographics, diet (formula versus human milk), and antibiotic use for the participating infants needs to be provided to give the appropriate context for interpretation of the research results.

Response to the reviewer comment: Done; All newborn babies' subjects in this study were chosen with normal breast feeding. Further future studies for comparing breast and formula feeding effect on the postnatal initial microbiome colonizing the neonates' colon would be of interest. This was outlined in the limitations paragraph in revised form of the manuscript.

Also, all neonates' subjects in this study were chosen healthy and received no special health care. It would be of interest in future studies to investigate the effect of various health care conditions and antimicrobial medications on postnatal initial bacterial

colonization of colon in health care subjected neonates such as neonates of preterm birth in comparison to healthy newborn babies. This was outlined in the limitations paragraph in revised form of the manuscript.

What is the standard of care for c-section deliveries at the hospital? Which antibiotics are used? How long could they be expected to be in the bloodstream of the mother and infant? Could that regimen affect the viability of microbes in the infant?

Response to the reviewer comment: Done; Despite the concern about the influence of antibiotic exposure of healthy mothers prior to labor or operative period for C-section on the neonates' meconium microbiota, the results in this study showed no *E. coli* colonization in also first-pass meconium of healthy newborn babies of vaginal delivery confirming a postnatal colonization. Further future studies would be of interest to investigate the effect of antibiotic exposure of health care requiring mothers prior to vaginal or C-section delivery on the neonates' colon postnatal bacterial colonization. This was outlined in the revised form of the manuscript.

Detailed information about sample collection is required in the methods section. Were the samples collected from diapers? Or, was sampling done using swabs? By whom and when were the samples collected? Were samples stored at all before processing? If so, how?

Response to the reviewer comment: Done; As mentioned above Once first-pass meconium collected in first day after birth or subsequent days stool samples were obtained normally in diapers, they were sent to the bacteriology laboratory and directly used for examining the occurrence of *E. coli* and determining its CFU. This was outlined in the revised form of the manuscript.

Was regular *Taq* (line 135) not high fidelity polymerase used for preparation of the 16S genes for sequencing? If so, this could introduce errors in the sequences and should be listed as a limitation. This is especially true in light of the length of PCR product generated.

Response to the reviewer comment: Done; In this study, regular *Taq* polymerase was used. Although high fidelity polymerases, is usually used in amplification for molecular cloning, it also can be preferably used in future studies in amplification and

sequencing of 16S rRNA for phylogenetic analysis. This was outlined in the revised form of the manuscript.

With respect to the collection of colonies for determination of *E. coli* strains, please clarify if a single colony per infant was selected or how many colonies were sampled per infant for the sequencing.

Response to the reviewer comment: Done; From the bacterial colonies identified by morphological and biochemical characterizations for stool sample of each infant, a single bacterial isolate colony per infant was selected from each of the healthy babies' subjects for further identification by phylogenetic analysis of 16S rRNA encoding gene sequence. This was outlined in the revised form of the manuscript.

The discussion section (and results section) could be reduced in length by removing the repeated statement of the same results that can found in many, many locations throughout the manuscript.

Response to the reviewer comment: Done, revised; The following and others were removed from the discussion section where it was clear enough in the results section (Despite the early appearance of *E. coli* in the stool of normal birth babies in comparison to that of newborn babies of cesarean section surgical delivery, there were comparable CFU mean numbers of the bacterium in both mode of delivery in 5 and 6 days old babies where there was a sharper colonization for newborn babies of cesarean section surgical delivery upon inoculation indicating better suitability of the condition for colonization at this age. This also can be attributed to a more space for bacterial colonization in the newborn babies of cesarean section surgical delivery that was not inoculated yet by *E. coli* and possibly other bacteria due to the more sterile delivery mode of cesarean section surgery away from the bacterial flora of the vaginal canal.)

If you have not cited the work of Andreas Baumler https://health.ucdavis.edu/medmicro/Faculty_MR/Baumler/baumler_index_mr.html in your discussion of the de-oxygenation of the gut and the role that plays in establishment of the gut microbiota, then please do.

Response to the reviewer comment: Done; The following was outlined in the discussion section of the revised form of the manuscript:

Understanding the interactions among the microbiota, the host and pathogenic bacteria can help in exploring strategies to manipulate the gut microbiota against enteric pathogens (Bäumler and Sperandio, 2016). Interestingly, the commensal *Enterobacteriaceae E. coli* was found to protect against neonatal chicken intestinal colonization by pathogenic *Salmonella* through their competition for oxygen consumption which reduces expansion of this pathogen that occur under aerobic respiration conditions (Litvak, *et al.*, 2019).

Bäumler, A. J., and Sperandio, V. (2016) Interactions between the microbiota and pathogenic bacteria in the gut. *Nature*, 535(7610), 85-93.

Litvak, Y., Mon, K., Nguyen, H., Chanthavixay, G., Liou, M., Velazquez, E. M., Kutter, L., Alcantara, M. A., Byndloss, M. X., Tiffany, C. R., Walker, G. T., Faber, F., Zhu, Y., Bronner, D. N., Byndloss, A. J., Tsolis, R. M., Zhou, H., and Bäumler, A. J. (2019). Commensal *Enterobacteriaceae* protect against *Salmonella* colonization through oxygen competition. *Cell Host Microbe*. 25(1), 128-139.e5.

In several places, it is suggested that the vagina might be the source of the *E. coli*. However, it is more likely that fecal material would be the source of *E. coli*.

Response to the reviewer comment: Done; As *E. coli* appeared slightly earlier in vaginal birth neonates than C-section ones, the vagina might be the source of inoculation of the neonates' gut by this bacterium during vaginal delivery. This was outlined clear in the revised form of the manuscript

Consider using the word "vaginal" instead of "normal" with reference to birth.

Response to the reviewer comment: Done.

Consider using the word, "sex" instead of the word, "gender."

Response to the reviewer comment: Done.

September 19, 2021

Prof. Fatthy M Morsy
Taibah University
Biology
Elsalam
Al-Madinah, Al-Madinah
Saudi Arabia

Re: Spectrum00379-21R1 (Prenatal versus postnatal initial colonization of healthy neonates' colon ecosystem by the enterobacterium Escherichia coli)

Dear Prof. Fatthy M Morsy:

Thank you for submitting your manuscript to Microbiology Spectrum. When submitting the revised version of your paper, please provide (1) point-by-point responses to the issues raised by the reviewers as file type "Response to Reviewers," not in your cover letter, and (2) a PDF file that indicates the changes from the original submission (by highlighting or underlining the changes) as file type "Marked Up Manuscript - For Review Only". Please use this link to submit your revised manuscript - we strongly recommend that you submit your paper within the next 60 days or reach out to me. Detailed information on submitting your revised paper are below.

Link Not Available

Sincerely,

Laxmi Yeruva

Journals Department
Reviewer comments:

Reviewer #1 (Comments for the Author):

Authors covered all points raised by the reviewer.

Reviewer #2 (Comments for the Author):

Thank you for addressing most of my comments. However, there are still issues with this manuscript.

For instance, in lines 203-205 and again in lines 275-280, you mention the vagina as the original source of the E. Coli. However, during vaginal birth infants are exposed to maternal fecal matter -- babies mouths actually are oriented toward the maternal rectum. Most women will poop a little during vaginal birth. Thus, this fecal matter may be the source of the E. Coli.

The phrase "healthy babies" still appears in this manuscript, but it seems that some of the infants may not have been healthy if I am interpreting your methods section correctly.

Furthermore, even with edits, bullets 5&6 (the last two) in "highlights" are not appropriate given the results reported in the manuscript.

Staff Comments:

Preparing Revision Guidelines

Please return the manuscript within 60 days; if you cannot complete the modification within this time period, please contact me. If you do not wish to modify the manuscript and prefer to submit it to another journal, please notify me of your decision immediately so that the manuscript may be formally withdrawn from consideration by Microbiology Spectrum.

Point-to-Point Response to Reviewers

Manuscript ID: Spectrum00379-21

The authors are grateful to the valuable comments of the reviewer. The manuscript has been revised according to the reviewer's comments.

Response to Reviewer #2

The authors are grateful to the valuable comments of the reviewer.

Reviewer #2 (Comments for the Author):

Thank you for addressing most of my comments. However, there are still issues with this manuscript.

For instance, in lines 203-205 and again in lines 275-280, you mention the vagina as the original source of the E. Coli. However, during vaginal birth infants are exposed to maternal fecal matter -- babies' mouths actually are oriented toward the maternal rectum. Most women will poop a little during vaginal birth. Thus, this fecal matter may be the source of the E. Coli.

Response to the reviewer comment: Done; Revised. Inoculation of the babies' gut by bacteria can thus occur during the babies' pass in the vaginal canal and/or from surrounding birth environment just after release possibly from residual contaminations with maternal fecal matter where most women poop a little during vaginal birth. This was outlined in the revised form of the manuscript.

The phrase "healthy babies" still appears in this manuscript, but it seems that some of the infants may not have been healthy if I am interpreting your methods section correctly.

Response to the reviewer comment: Done; All neonates were selected healthy. This was the basic condition for selecting neonates in this study. This was outlined in the revised form of the manuscript.

Furthermore, even with edits, bullets 5&6 (the last two) in "highlights" are not appropriate given the results reported in the manuscript.

Response to the reviewer comment: Done; highlights were revised to be concise.

September 23, 2021

Prof. Fatthy M Morsy
Taibah University
Biology
Elsalam
Al-Madinah, Al-Madinah
Saudi Arabia

Re: Spectrum00379-21R2 (Prenatal versus postnatal initial colonization of healthy neonates' colon ecosystem by the enterobacterium Escherichia coli)

Dear Prof. Fatthy M Morsy:

Your manuscript has been accepted, and I am forwarding it to the ASM Journals Department for publication. You will be notified when your proofs are ready to be viewed.

Sincerely,

Laxmi Yeruva
Editor, Microbiology Spectrum
